# Effect of Methyl Jasmonate and Methyl Jasmonate Plus Urea Foliar Applications on Wine Phenolic, Aromatic and Nitrogen Composition

Miriam González-Lázaro, Itziar Sáenz de Urturi, Rebeca Murillo-Peña, Sandra Marín-San Román, Eva P. Pérez-Álvarez *, Pilar Rubio-Bretón and Teresa Garde-Cerdán *

Grupo VIENAP, Instituto de Ciencias de la Vid y del Vino (CSIC, Universidad de La Rioja, Gobierno de La Rioja), Ctra. de Burgos, km. 6, 26007 Logroño, Spain

* Correspondence: evapilar.perez@icvv.es (E.P.P.-Á.); teresa.garde.cerdan@csic.es (T.G.-C.)

**Abstract:** Foliar application has been studied to enhance grape composition and, therefore, wine quality. This work examined, for first time, the effects of foliar applications of methyl jasmonate (MeJ) and methyl jasmonate plus urea (MeJ+Ur) to Tempranillo vineyard on wine phenolic, aromatic and nitrogen composition over two vintages (2019 and 2020). A reduction in alcoholic degree was observed in MeJ and MeJ+Ur wines. The effect of foliar treatments was season-dependent. MeJ and MeJ+Ur wines were characterized, in the first vintage, by a higher content of total acylated anthocyanins, but a low content of total esters, alcohols and acids when compared with control wines. MeJ+Ur wines presented a higher total amino acids content than control and MeJ wines. However, in the second vintage, MeJ and MeJ+Ur wines presented an increase in some non-acylated anthocyanins, but only MeJ+Ur treatment increased the total content of flavonols, flavanols, hydroxycinnamic acids, stilbenes and total amino acids when compared with control wines. MeJ wines presented a low content of esters and acids, whereas MeJ+Ur did not show differences with control wines. Overall, the effect of MeJ+Ur foliar treatment was greater than the effect of MeJ application in order to improve the wine chemical composition.

**Keywords:** phenolic compounds; volatile compounds; amino acids; urea; methyl jasmonate; red wine

## 1. Introduction

Phenolic, volatile and nitrogen compounds are key to wine quality. On one hand, phenolic compounds include a mixed group of compounds that are formed across the phenylpropanoid pathway, which precursor is the phenylalanine [1]. Phenolics are related to wine color, mouthfeel properties and the ageing potential of wines [2]. In addition, these compounds are associated with beneficial health properties [3]. On the other hand, nitrogen compounds affect the development of alcoholic fermentation and the formation of flavour metabolites, which can be controlled by the addition of nitrogen into the vineyard or in the winery [4–6]. Finally, wine aroma is influenced by hundreds of volatile compounds, which belong to different chemical families of compounds that are grouped in different classes: varietal aroma, pre-fermentative aroma, fermentative aroma and post-fermentative aroma [7]. Fermentative aromas represent, quantitatively, the wine aroma, and among these volatile compounds, esters, higher alcohols and acids are the main ones responsible for the fermentation bouquet [8].

To improve wine's quality, several approaches have been studied: agronomical practices in the vineyard [9], and innovative techniques in the cellar [10,11]. In this sense, the foliar application of elicitors and nitrogen compounds to vineyard has been studied to palliate the effects of climate change in grape content, and therefore, to enhance grape and wine quality. MeJ is an elicitor, i.e., a compound able to trigger a response of defense in plants that induce the production of secondary metabolites, such as phenolic and volatile

compounds [12–14]. Previous works showed that foliar treatments carried out with MeJ increased the anthocyanin, flavonol and stilbene content in wines [15,16]; on the other hand, an increase in certain anthocyanins and flavonols was also described after urea foliar application [17]. It is noteworthy that the stimulus of phenolic compounds synthesis seems to be dependent on the time of application of urea, since at veraison, the vegetative growth of plant was slow; therefore, enhanced grapevine metabolism induces the accumulation of secondary metabolites [18]. Garde-Cerdán et al. [19] also observed an increase in resveratrol and piceid content in must and wines after urea foliar application to vines. However, a greater effect of MeJ foliar application in the phenolic content of grapes was observed than those reflected in wines [20], probably due to the effect of elicitors in the cell wall structure. Paladines-Quezada et al. [21] showed that the spray application of MeJ on vine clusters produced changes in certain components of the skin cell walls, which can reinforce the skin cell walls, doing more difficult the anthocyanin extraction during the winemaking. This reinforcement of the skin cell wall was also observed by Apolinar et al. [22] after foliar application of MeJ. There are few papers about the effect of MeJ and urea foliar application on wine amino acids content. Gil-Muñoz et al. [23] described an increase in the content of several amino acids in must and wine, although they also observed a climatic dependence on the effect of MeJ treatment. Finally, the effect of MeJ and urea foliar application, separately, on volatile compounds in wines has been described in the Bibliography. Regarding the effect of MeJ foliar application to vineyard, Gómez-Plaza et al. [13] described a higher content of terpenes and norisoprenoids in wines elaborated with MeJ grapes. D'Onofrio et al. [14] also observed an increase in monoterpenes in grapes and wines, with an impact on wine aroma longevity and the sensorial characters of wines. On the other hand, Ancín-Azpilcueta et al. [24] showed that urea foliar application improved wine aroma, since the total content of alcohols decreased and some esters (ethyl hexanoate, ethyl octanoate, and ethyl decanoate) increased. However, Rubio-Bretón et al. [25] concluded that urea foliar application did not modify the wine aroma profile. Hence, the effect of combined application of MeJ and urea could be a good strategy to improve wine quality and, as far as we know, their combined foliar application has never been studied previously. Therefore, the aim of this work was to study, for the first time, the effects of foliar treatments of MeJ and MeJ+Ur to Tempranillo vineyard on wine phenolic, aromatic and nitrogen composition over two vintages.

## 2. Materials and Methods

### 2.1. Vineyard Site, Grapevine Treatments and Vinification

This trial was conducted during two growing seasons (2019 and 2020) with red grapes from Tempranillo (*Vitis vinifera* L.) variety grown in the experimental vineyard of Finca La Grajera, located in Logroño, La Rioja (Spain) (Lat: 42°26′25.36″ North; Long: 2°30′56.41″ West; 456 m above sea level). All vines were planted in 1997, grafted onto a R-110 rootstock and were trained to a VSP (vertical shoot positioned) trellis system with a grapevine spacing of 2.80 m × 1.25 m. Three treatments were carried out: (i) control, (ii) methyl jasmonate (MeJ), and (iii) methyl jasmonate plus urea (MeJ+Ur).

All the products were dissolved in water, with a concentration of 10 mM of methyl jasmonate (MeJ) (following previous works, Garde-Cerdán et al. [26,27] and the equal concentration of methyl jasmonate plus urea in a dose of 6 kg N/ha (MeJ+Ur) (according to exposed by Pérez-Álvarez et al. [28]). Tween 80 (1 mL/L) was used as a wetting agent. Control plants were sprayed with an aqueous solution of Tween 80 alone. Treatments were performed at the vineyard twice, at veraison and one week later. In total, 200 mL of each solution was sprayed over leaves. The treatments were carried out in triplicate and the experimental design was arranged in a complete randomized block design along the vineyard, with 10 vines for each replication and treatment.

The grapes from all grapevines and treatments were manually harvested at their optimum technological maturity, i.e., when the weight of 100 berries remained constant and the probable alcohol reached 13 (% *v/v*). At the winery, the grape clusters were destemmed and

crushed, grapes from each treatment and replicate were elaborated separately. The resulted paste–must was introduced in one 30 L-tank for each one. Therefore, 9 elaborations were performed (3 treatments × 3 repetitions/treatment). They were protected by the addition of 50 mg $SO_2$/kg of grapes and the alcoholic fermentation was induced by inoculation (at a dosage of 20 g/hL) with a commercial *Saccharomyces cerevisiae* strain (Safoeno SC22, Fermentis, Marcq-en-Barœul, France). The fermentation was carried out at a controlled temperature of 20 +/− 2 °C. Once the alcoholic fermentation finished (the must values of residual sugars were below than 2.5 g/L), a comercial *Oenococcus oeni* strain (Viniflora CiNe, CHR Hansen, Hørsholm, Denmark) at 1 g/hL was inoculated into the wines, in order to carry out the malolactic fermentation (MLF) (at 17 +/− 1° C). Once the MLF was finished, the enological parameters were analyzed, and, for each wine and for each group of compounds studied (phenolic compounds, volatile compounds and amino acids), aliquot samples were frozen and stored at −20 °C until their analysis.

### 2.2. Determination of Enological Parameters

Wines were characterized by measuring the alcoholic degree, pH, total acidity, volatile acidity, total anthocyanins, color index (CI) and total polyphenol index (TPI) using the official methods established by the OIV [29]. Malic acid, lactic acid, yeast assimilable nitrogen (YAN) and total phenols were determined using a Miura One enzymatic equipment (TDI, Barcelona, Spain).

As field treatments were performed in triplicate and one vinification was carried out from each replicate, the results of wine enological parameters are shown as the average of three samples (*n* = 3).

### 2.3. Analysis of Wine Phenolic Compounds by HPLC-DAD

2.3.1. Sample Preparation for the Determination of Non-Anthocyanin Phenolic Compounds

To avoid interferences in the chromatographic separation and identification of anthocyanins, an extraction of non-anthocyanin phenolic compounds was performed, according to Portu et al. [18]. PCX SPE cartridges (500 mg, 6 mL; Bond Elut Plexa, Agilent, Palo Alto, CA, USA) were used. Cartridges were located in the extraction system (VisiprepTM Vacuum Manifold, Sigma-Aldrich, Madrid, Spain). Firstly, wines (3 mL) were diluted with 3 mL of 0.1 N HCl. The PCX SPE cartridges were conditioned with 5 mL of methanol and 5 mL of water. Then, the diluted samples were passed across the PCX SPE cartridges and washing step was performed with 5 mL of 0.1 N HCl and 5 mL of water. The non-anthocyanin phenolic compounds fraction was eluted with 2 × 3 mL of ethanol, and then was dried in a centrifugal evaporator (miVac, Genevac Ltd., Suffolk, UK) at 35 °C and re-solved in 1.5 mL of 20 % (*v/v*) methanol aqueous solution. The anthocyanin-free fraction was used to analyze non-anthocyanin phenolic compounds (flavonols, flavanols, hydroxybenzoic and hydroxycinnamic acids and stilbenes).

2.3.2. Analysis of Phenolic Compounds by HPLC-DAD

Wine phenolic compounds were analyzed by reverse-phase HPLC, according to Portu et al. [18], using an Agilent 1260 Infinity II chromatograph, equipped with a diode array detector (DAD). Wine samples were filtered and directly injected on a Licrospher® 100 RP-18 reversed-phase column (250 × 4.0 mm; 5 μm packing; Agilent) with pre-column Licrospher® 100 RP-18 (4 × 4 mm; 5 μm packing; Agilent), both thermostated at 40 °C, for the analysis of anthocyanins. A flow rate of 0.630 mL/min was established and 10 μL of wine was injected. The eluents used included (A) acetonitrile/water/formic acid (3:88.5:8.5, *v/v/v*), and (B) acetonitrile/water/formic acid (50:41.5:8.5, *v/v/v*). For the analysis of non-anthocyanin phenolic compound fractions, the injection volume was 20 μL. The same column was used, whereas the eluents were (A) acetonitrile/water/formic acid (3:88.5:8.5, *v/v/v*), (B) acetonitrile/water/formic acid (50:41.5:8.5, *v/v/v*), and (C) methanol/water/formic acid (90:1.5:8.5, *v/v/v*).

Phenolic compounds were identified according to the retention times of available pure compounds and the UV-Vis data obtained from authentic standards and/or published in previous studies [30]. Quantification was performed with DAD chromatograms, and were extracted at 520 nm (anthocyanins), 360 nm (flavonols), 320 nm (hydroxycinnamic acids and stilbenes), and 280 nm (gallic acid and flavanols) and the calibration graphs of the respective standards ($R^2 > 0.99$) were used. When a standard was not available, quantification was carried out according to the calibration graph of the most similar compound. Hence, malvidin-3-*O*-glucoside (Extrasynthèse, Genay, France) was used for anthocyanins, quercetin-3-*O*-glucoside (Sigma-Aldrich) was used for flavonols, gallic acid was quantified with gallic acid (Sigma-Aldrich), *trans*-caftaric acid (Extrasynthèse) was used for free hydroxycinnamic acids and the corresponding tartaric esters, catechin (Sigma-Aldrich) was used for procyanidins B1 and B2, epicatechin (Sigma-Aldrich) was used for epigallocatechin, and *trans*-piceid (Sigma-Aldrich) and *trans*-resveratrol (Sigma-Aldrich) were used for their respective *cis* isomers. Concentrations of phenolic compounds in wines were expressed as mg/L. The characteristics of the HPLC method were the following: variation coefficient (%) for retention time of commercially standards varied from 0.09 to 0.72; the detection limit (mg/L) ranged from 0.099 to 0.711; the quantification limit (mg/L) changed from 0.292 to 2.370; the variation coefficient (%) for concentration varied from 1.66 to 6.67. The response factor (mg/area units) was also calculated ranging from $3.99 \times 10^{-6}$ to $1.00 \times 10^{-4}$. The variation coefficients were obtained from 10 consecutive analyses.

Since treatments were carried out in triplicate and one vinification was performed from each replicate, the results for phenolic compounds are the average of the analyses of three samples ($n = 3$).

### 2.4. Determination of Wine Aromatic Compounds by GC-MS

Wine volatile compounds were analyzed following the method described by Garde-Cerdán et al. [8]. In a tube of 10 mL, 8 mL of wine (previously centrifuged at $3220 \times g$, during 15 min, at 4 °C), 10 µL of internal standard (2-octanol, Sigma-Aldrich) and a magnetic stir bar were added. The extraction of wine volatile compounds was carried out by stirring the sample during 15 min, with 400 µL of dichloromethane (Merck, Darmstadt, Germany). After cooling the samples for 10 min at 0 °C, the organic phase was separated by centrifugation ($5031 \times g$, 10 min, 4 °C). Then, the extract was recovered into a vial. Gas chromatographic determination of volatile analytes was performed using a Gas Chromatograph (GC) with a Mass Detector (MS) (Agilent). In total, 2 µL of extract was injected in split mode. A VF-Wax 52 CB (60 m × 0.25 mm i.d. × 0.25 µm) capillary column (Agilent) was utilized. The injector temperature was programmed from 40 °C to 250 °C, at 180 °C/min. The temperature of the oven was maintained during 2 min at 50 °C, then a rise was programmed at 3 °C/min from 50 °C to 250 °C. The detector run at electronic impact mode (70 eV), with an acquisition range (m/z) from 29 to 260. The identification of volatile compounds was performed using the NIST library and by comparison with the mass spectrum of available standards (Sigma-Aldrich). A semi-quantification was carried out, relating the areas of each volatile compound with the area and the known concentration of the internal standard. For the validation of the GC-MS method, these characteristics were obtained: limit of quantification (µg/L) was ranged from 1.8 to 107.4; the concentration range (µg/L) varied from 2.48 to 2132; precision was expressed as relative standard deviation (RSD, %) and changed from 5.7 to 19.7; the repeatability (%) was measured from 2.6 to 9.0; the accuracy was expressed as relative error (%) and varied from 0.7 to 20.5.

As the treatments were performed in triplicate, and one vinification was performed from each replicate, the results of wine volatile compounds are shown as the average of three samples ($n = 3$).

### 2.5. Analysis of Wine Nitrogen Compounds by HPLC-DAD

The determination of amino acids was carried out following the method described by Garde-Cerdán et al. [31]. In short, amino acids were derivatized by the reaction of

1.75 mL of borate buffer 1 M (pH 9), 750 μL of methanol (Merck), 1 mL of wine (previously filtered), 20 μL of internal standard (2-aminoadipic acid, 1 g/L) (Sigma-Aldrich) and 30 μL of derivatization reagent diethyl ethoxymethylenemalonate (DEEMM) (Sigma-Aldrich). The reaction of derivatization was carried out in a screw-cap test tube during 30 min in an ultrasound bath. Then, samples were heated at 70–80 °C for 2 h to allow for the complete degradation of excess DEEMM and reagent by-products.

The analyses were performed on a Shimadzu Nexera X2 ultra high-performance liquid chromatograph equipment (Shimadzu, Kyoto, Japan) with a photodiode array detector (DAD). Chromatographic separation was carried out in an ACE HPLC column (C18-HL) (Aberdeen, Scotland) particle size 5 μm (250 mm × 4.6 mm). The elution of amino acids was performed under the conditions described by Garde-Cerdán et al. [32]. Phase A, 25 mM acetate buffer, pH 5.8, with 0.4 g of sodium azide; phase B, 80:20 (*v/v*) mixture of acetonitrile and methanol (Merck). Amino acids were detected by DAD monitored at 280, 269 and 300 nm. 50 μL of sample were the injection volume. Identification of target compounds were performed according to the retention times and the UV–Vis spectral characteristics of the corresponding derivatized standards (aspartic acid, glutamic acid, asparagine, serine, glutamine, histidine, glycine, threonine, citrulline, arginine, alanine, γ-aminobutyric acid, proline, tyrosine, valine, methionine, cysteine, isoleucine, tryptophan, leucine, phenylalanine, ornithine, and lysine, all from Sigma-Aldrich). Quantification was carried out by using the calibration graphs of the respective standards in 0.1 N HCl ($R^2 > 0.96$), which underwent the same process of derivatization that the samples. The characteristics measured in the validation of the method were: range of calibration (mg/L) from 1.00 to 2484; the repeatability of the method was studied, and the resulting variation coefficients were below 5%; the detection limits for the amino acids were below 0.4 mg/L.

The treatments in vineyard were performed in triplicate, as were the vinifications; thus, the results of free amino acids correspond to the average of 3 analyses ($n = 3$).

### 2.6. Statistical Analysis

The statistical elaboration of the data was performed using SPSS Version 21.0 statistical package for Windows (SPSS, Chicago, IL, USA). General parameters and phenolic, aromatic and nitrogen compounds data were processed using the variance analysis (ANOVA) ($p \leq 0.05$). The differences between means were compared using Duncan's test.

## 3. Results and Discussion

### 3.1. Effect of MeJ and MeJ+Ur Foliar Applications on Wine Enological Parameters

All wines showed usual values of enological parameters for Tempranillo wines from La Rioja region (Table 1) [15,18].

In the first season (2019), wines presented significant differences among control wines and wines made from treated grapevines (MeJ and MeJ+Ur), with regard to alcoholic degree, which is interesting to palliate the effect of climatic change in grapevines, since both treatments produced a decrease in the alcoholic degree of wines.

This effect can be explained because MeJ treatment can accelerate or delay the grapes ripening when is applied at low or high concentrations [14,33]. Moreover, Paladines-Quezada et al. [21] described a decrease in Brix in Merlot grapes from MeJ treatment, supporting the hypothesis that MeJ treatment can delay grape maturation. Furthermore, Ancín-Azpilicueta et al. [24] described a decrease in the alcoholic strength and total acidity in wines elaborated with grapes from grapevines foliar treated with urea. Both treatments, MeJ and MeJ+Ur, decreased the total acidity of wines when compared with control wines (Table 1). MeJ wines showed the highest volatile acidity, but values are correct for a good process of fermentation. It is noteworthy that, in this season (2019), both treatments produced wines with a higher YAN than control wines, being higher the YAN content in MeJ+Ur wines than in MeJ wines. This effect has been observed in Tempranillo grapes by Garde-Cerdán at al. [6], who described an increase in YAN content after urea foliar application when compared with control grapes. Nevertheless, for MeJ foliar treatment,

Gutiérrez-Gamboa et al. [34] showed an absence of effect of this elicitor on YAN content in Tempranillo grapes. In the second season (2020), wines showed slight differences among control wines and wines made from treated grapevines (Table 1). MeJ wines presented the lowest volatile acidiy and control wines presented the lowest lactic acid content. With regard to the rest of the enological parameters, differences among wines were not found.

**Table 1.** General parameters in wines from control, methyl jasmonate (MeJ) and MeJ+Urea (MeJ+Ur) treatments, in 2019 and 2020.

| | 2019 | | | 2020 | | |
| | Control | MeJ | MeJ+Ur | Control | MeJ | MeJ+Ur |
|---|---|---|---|---|---|---|
| **Alcohol (% *v/v*)** | 13.97 ± 0.31 b | 12.57 ± 0.25 a | 12.80 ± 0.40 a | 12.47 ± 0.70 a | 12.18 ± 1.59 a | 12.53 ± 0.81 a |
| **pH** | 3.96 ± 0.07 a | 3.90 ± 0.10 a | 3.94 ± 0.13 a | 3.66 ± 0.08 a | 3.70 ± 0.04 a | 3.73 ± 0.13 a |
| **Total acidity (g/L) *** | 4.27 ± 0.10 b | 4.08 ± 0.06 a | 3.92 ± 0.06 a | 4.43 ± 0.59 a | 4.38 ± 0.23 a | 4.02 ± 0.23 a |
| **V A [1] (g/L) **** | 0.23 ± 0.02 a | 0.28 ± 0.03 b | 0.20 ± 0.01 a | 0.22 ± 0.02 b | 0.18 ± 0.01 a | 0.22 ± 0.03 b |
| **Lactic acid (g/L)** | 1.32 ± 0.10 a | 1.36 ± 0.07 a | 1.28 ± 0.12 a | 0.86 ± 0.07 a | 1.14 ± 0.15 b | 1.05 ± 0.09 b |
| **YAN [2] (mg N/L)** | 18.06 ± 2.08 a | 41.65 ± 3.90 b | 67.89 ± 8.90 c | 30.36 ± 0.54 a | 28.40 ± 12.49 a | 39.34 ± 10.65 a |
| **T P [3] (mg/L)** | 2440.83 ± 123.16 a | 2160.37 ± 221.12 a | 2460.73 ± 124.74 a | 1116.63 ± 106.69 a | 1263.07 ± 224.95 a | 1333.47 ± 153.38 a |
| **T A [4] (mg/L)** | 1117.33 ± 69.97 a | 1225.67 ± 98.64 a | 1289.67 ± 102.00 a | 130.99 ± 20.13 a | 158.53 ± 18.35 a | 168.00 ± 18.68 a |
| **CI [5]** | 18.27 ± 1.03 a | 17.53 ± 1.81 a | 19.01 ± 1.14 a | 6.05 ± 0.55 a | 7.70 ± 2.13 a | 8.62 ± 1.10 a |
| **TPI [6]** | 70.83 ± 3.47 a | 66.43 ± 7.95 a | 73.32 ± 5.00 a | 36.82 ± 4.05 a | 41.04 ± 8.69 a | 44.73 ± 5.62 a |

[1] V A: Volatile acidity, [2] YAN: yeast assimilable nitrogen, [3] T P: Total phenols, [4] T A: Total anthocyanins, [5] CI: Color index, [6] TPI: Total polyphenol index. * As g/L tartaric acid. ** As g/L acetic acid. All parameters are listed with their standard deviation ($n = 3$). For each season and compound, different letters indicate significant differences between the samples ($p \leq 0.05$).

With regard to the enological parameters related to wine color, i.e., total phenols, total anthocyanins, CI and TPI, did not find differences among control wines and wines made from treated grapevines (MeJ and MeJ+Ur) at any of the two seasons studied. This result was in contrast to the results detailed by Portu et al. [1], who indicated that MeJ treatment enhanced the wine chromatic parameters; moreover, Portu et al. [18] described an increase in total anthocyanins and TPI in wine samples from grapevines treated with urea with respect to control wines.

*3.2. Influence of the Foliar MeJ and MeJ+Ur Treatments on Wine Phenolic Composition*

Table 2 shows the results of anthocyanins content in control wines and wines elaborated with grapes from grapevines treated (MeJ, and MeJ+Ur). Non-acylated anthocyanins were the main contributors to the total anthocyanins content and malvidin-3-glc was the major anthocyanin in all wines in both seasons. In the first season, MeJ and MeJ+Ur wines showed an increase in several anthocyanins and in total acylated anthocyanins when compared with control wines, although MeJ+Ur wines did not show differences in total acylated anthocyanins with control wines. MeJ wines presented a higher content of peonidin-3-glc, cyanidin-3-acglc, peonidin-3-acglc, cyanidin-3-cmglc, petunidin-3-cmglc, peonidin-3-cmglc, malvidin-3-*trans*-cmglc, and total acylated anthocyanin, and a lower content of vitisin A when compared with control wines (Table 2).

For its part, MeJ+Ur wines showed a higher content of peonidin-3-glc, cyanidin-3-acglc, cyanidin-3-cmglc, and peonidin-3-cmglc in comparison with control wines and also, a lower content of vitisin A. The main differences among wines from grapevines treated were that MeJ+Ur wines showed a lower content of malvidin-3-*trans*-cmglc and malvidin-3-cfglc than MeJ wines (Table 2).

The increase observed in the total acylated anthocyanins is interesting because this kind of anthocyanins are more stables for wine color. Moreover, this effect has been observed previously in grapes from vine foliar-treated with MeJ or urea. Portu et al. [17] showed that the foliar application of urea to vineyards induced anthocyanin synthesis and therefore this nitrogen treatment is a good tool in order to improve the anthocyanin content of grapes and thus grape quality. On the other hand, MeJ is an elicitor able to trigger the activation of enzymes involved in the synthesis of phenolic compounds [12,35].

However, the effect of both foliar treatments in grape composition was higher that thus observed in wine composition, since MeJ and MeJ+Ur produced an increase in several anthocyanins and total anthocyanins in grapes (in the first season studied, data pending publication). This result could be explained for the following reasons: anthocyanins are located in the grape skins, and inside the cells, anthocyanins are located in the vacuoles. During winemaking, they are extracted in the maceration process. To achieve this, the pectin rich middle lamella must be degraded to release the cells and then, anthocyanins can diffuse into the wine [36]. The fact that the differences found in grapes were minimized throughout the fermentation process could be due to a lower release or diffusion of phenolic compounds during winemaking. This idea makes sense since Paladines-Quezada et al. [21] found that MeJ treatment can increase protein concentration in the skin cell wall of grapes and this produces a more rigid cell wall [1,36]. Moreover, it should be noted that the foliar treatments were performed at veraison and one week later, when anthocyanins are beginning to accumulate. Most likely, the effects of the treatments are conditioned by the time of application [18].

**Table 2.** Anthocyanins content (mg/L) in wines from control, methyl jasmonate (MeJ) and MeJ+Urea (MeJ+Ur) treatments, in 2019 and 2020 seasons.

| | 2019 | | | 2020 | | |
| | Control | MeJ | MeJ+Ur | Control | MeJ | MeJ+Ur |
|---|---|---|---|---|---|---|
| Delphinidin-3-glc | 14.67 ± 2.72 a | 17.06 ± 1.23 a | 15.68 ± 1.44 a | 6.48 ± 0.67 a | 11.03 ± 1.09 b | 10.20 ± 1.88 b |
| Cyanidin-3-glc | 2.21 ± 0.06 a | 2.44 ± 0.41 a | 2.54 ± 0.33 a | 1.57 ± 0.07 a | 1.78 ± 0.19 ab | 1.88 ± 0.13 b |
| Petunidin-3-glc | 20.48 ± 3.40 a | 22.94 ± 3.45 a | 22.68 ± 1.06 a | 13.81 ± 2.37 a | 18.22 ± 1.49 b | 17.45 ± 2.27 ab |
| Peonidin-3-glc | 6.38 ± 0.60 a | 9.43 ± 0.84 b | 8.70 ± 1.07 b | 2.83 ± 0.56 a | 4.11 ± 0.55 ab | 4.33 ± 0.90 b |
| Malvidin-3-glc | 89.68 ± 8.97 a | 101.81 ± 5.10 a | 99.17 ± 3.46 a | 82.84 ± 8.04 a | 80.27 ± 17.19 a | 89.45 ± 8.15 a |
| Total non-acylated | 133.42 ± 15.69 a | 153.68 ± 9.56 a | 148.77 ± 4.45 a | 107.53 ± 11.53 a | 115.40 ± 18.82 a | 123.29 ± 11.78 a |
| Delphinidin-3-acglc | 2.51 ± 0.24 a | 2.68 ± 0.13 a | 2.66 ± 0.07 a | 2.39 ± 0.19 a | 2.48 ± 0.38 a | 2.64 ± 0.16 a |
| Cyanidin-3-acglc | 1.35 ± 0.00 a | 1.37 ± 0.00 b | 1.36 ± 0.01 b | 1.36 ± 0.01 a | 1.37 ± 0.01 a | 1.38 ± 0.01 a |
| Petunidin-3-acglc | 2.61 ± 0.20 a | 2.67 ± 0.15 a | 2.66 ± 0.04 a | 2.59 ± 0.23 a | 2.64 ± 0.44 a | 2.77 ± 0.19 a |
| Peonidin-3-acglc | 2.12 ± 0.07 a | 2.60 ± 0.26 b | 2.41 ± 0.02 ab | 1.74 ± 0.10 a | 1.81 ± 0.17 a | 1.90 ± 0.09 a |
| Malvidin-3-acglc | 5.93 ± 0.46 a | 6.24 ± 0.09 a | 5.95 ± 0.19 a | 6.73 ± 0.44 a | 6.25 ± 0.94 a | 6.49 ± 0.30 a |
| Delphinidin-3-cmglc | 3.76 ± 0.35 a | 4.28 ± 0.37 a | 4.09 ± 0.13 a | 3.81 ± 0.57 a | 3.59 ± 0.68 a | 4.53 ± 0.58 a |
| Cyanidin-3-cmglc | 1.79 ± 0.09 a | 2.09 ± 0.17 b | 2.11 ± 0.09 b | 1.79 ± 0.11 a | 1.89 ± 0.29 a | 2.04 ± 0.15 a |
| Petunidin-3-cmglc | 2.90 ± 0.19 a | 3.30 ± 0.16 b | 3.11 ± 0.02 ab | 2.86 ± 0.35 a | 3.19 ± 0.45 a | 3.19 ± 0.38 a |
| Peonidin-3-cmglc | 2.37 ± 0.11 a | 2.91 ± 0.23 b | 2.82 ± 0.18 b | 2.28 ± 0.20 a | 2.44 ± 0.48 a | 2.66 ± 0.29 a |
| Malvidin-3-cis-cmglc | 1.71 ± 0.03 a | 1.74 ± 0.01 a | 1.70 ± 0.05 a | 1.82 ± 0.02 b | 1.70 ± 0.09 ab | 1.68 ± 0.06 a |
| Malvidin-3-trans-cmglc | 9.33 ± 0.46 a | 10.37 ± 0.38 b | 9.52 ± 0.33 a | 9.84 ± 1.52 a | 11.45 ± 2.60 a | 11.10 ± 2.30 a |
| Malvidin-3-cfglc | 1.99 ± 0.09 ab | 2.23 ± 0.17 b | 1.88 ± 0.17 a | 1.59 ± 0.06 a | 1.59 ± 0.26 a | 1.63 ± 0.08 a |
| Total acylated | 38.37 ± 2.22 a | 42.48 ± 0.97 b | 40.28 ± 0.45 ab | 38.80 ± 3.65 a | 40.41 ± 6.21 a | 42.01 ± 4.35 a |
| Total anthocyanins | 171.80 ± 17.75 a | 193.92 ± 14.13 a | 189.04 ± 4.81 a | 146.33 ± 15.18 a | 155.81 ± 24.83 a | 165.30 ± 16.01 a |
| Vitisin A | 2.00 ± 0.16 b | 1.73 ± 0.04 a | 1.68 ± 0.03 a | 1.51 ± 0.02 a | 1.53 ± 0.04 a | 1.55 ± 0.04 a |
| Vitisin B | 1.97 ± 0.12 a | 2.18 ± 0.18 a | 2.19 ± 0.05 a | 1.78 ± 0.05 a | 1.85 ± 0.23 a | 1.99 ± 0.08 a |

Nomenclature abbreviations: glc, glucoside; acglc, acetylglucoside; cmglc, trans-p-coumaroylglucoside; cfglc, caffeoylglucoside. All parameters are listed with their standard deviation ($n = 3$). For each season and compound, different letters indicate significant differences between the samples ($p \leq 0.05$).

In the second season studied (2020), the effect of the treatments on the final wines was minor. It is notable that the influence of foliar applications in the anthocyanin content of grapes in the second vintage also was minor, control and treated grapes did not show differences in total anthocyanin content (data pending to publish). MeJ wines showed a higher content of delphinidin-3-glc, and petunidin-3-glc, when compared with control wines, whereas MeJ+Ur wines showed a higher content of delphinidin-3-glc, cyanidin-3-glc, and peonidin-3-glc in comparison with control wines. Significant differences among wines from foliar-treated grapevines were not found (Table 2). These differences among the effect of the treatments according to season could be explained by differences in the pre-harvest rainfalls (Table S1). August in 2019 was drier than August in 2020, since weather differences among seasons could influence grape physicochemical composition and berry development and the response to the foliar application shows a meteorological dependence [21,37]. In view of these results, the maceration and extraction process carried out during winemaking

should be performed with extreme care to avoid losses of phenolic compounds and achieve the diffusion of these secondary metabolites that have been improved in grapes by foliar treatments carried out in the vineyard.

Table 3 shows the flavonol, flavanol, phenolic acid, and stilbene content in wines from control, MeJ and MeJ+Ur treatments in both seasons. In the first season, the main flavonol was myricetin-3-glc in all wines. Control wines were characterized by a higher content of myricetin-3-gal, quercetin-3-glcU, quercetin-3-glc, and kaempferol-3-glcU+3-glc than wines elaborated with grapes from grapevines treated. However, differences in the total flavonols content among wines were not found (Table 3). MeJ wines presented a higher content of free-quercetin, free-kaempferol, and free isorhamnetin+syringetin content than MeJ+Ur wines. Free flavonols are produced during the winemaking since flavonol aglycones are liberated by acid hydrolysis of the glycosides [16].

Furthermore, in the second season, the main flavonol was quercetin-3-glc. The effect of foliar treatments applied in vineyard was different, maybe due to the different range of concentration of phenolic compounds in this season owe to different meteorological conditions (Table S1), since the flavonols content in wines from 2020 was lower than those of 2019 vintage (Table 3), probably, due to a dilution effect on grapes content produced by the higher pre-harvest rainfalls measured in 2020 than in 2019 season (August). MeJ wines showed a higher content of quercetin-3-glc, isorhamnetin-3-glc, and free-myricetin than control wines. The effect of MeJ+Ur foliar treatment was greater than the effect of MeJ foliar treatment since, MeJ+Ur wines showed a higher content of all flavonols, except for quercetin-3glcU, kaempferol-3-gal, free-myricetin, free-laricitrin, and free-isorhamnetin+syringetin, than control wines. The total flavonols content of MeJ+Ur wines was higher than those of control wines but, did not show differences with regard to the total flavonols content of MeJ wines. These results showed that the effect of foliar treatments was season-dependent. Moreover, the increase observed in the flavonol content of wines from treated grapevines has a positive effect since these compounds are related to the color stability of wines.

Comparing with previous works, Portu et al. [1,16] described a non-effect on flavonol content in wines elaborated with grapevines treated with MeJ, whereas Portu et al. [15] showed an increase in total flavonol content in MeJ-treated wines. On the other hand, urea foliar application produced an increase in the concentration of certain flavonols in wines and in the total flavonols content, but only in one of the urea doses studied [18]. The different effect of foliar treatments on phenolic compounds can be due to climate conditions (Table S1) affect the response of vines to the treatments [21,38]. Potentially, the higher pre-harvest rainfalls recorded in 2020 than in 2019 vintage could affect the effect of foliar treatments. The differences observed among foliar treatments could be explained by a lower release of phenolic compounds during the winemaking owe to a greater consistency of the cell wall, since MeJ treatment can reinforce the skin cell wall of grapes, which hinders the extraction of phenolic compounds [21,22].

Flavanol content in wines is showed in Table 3. In the first season, MeJ wines were characterized by a higher content of epigallocatechin and procyanidin B1, but a lower content of procyanidin B2 when compared with control wines. MeJ+Ur wines showed a higher content of procyanidin B1 and B2 than control wines. These slight effects on flavanol content suggest that MeJ and MeJ+Ur treatments did not improve the synthesis of this family of flavonoids, as previously described by Portu et al. [15,16] for MeJ treatment, and Portu et al. [18] for urea foliar application. In the second vintage, the flavanol content of wines was lower than in 2019. Epicatechin-3-gallate and procyanidin B2 were not detected in 2020 (Table 3). The differences among control, MeJ and MeJ+Ur wines were slight. MeJ wines presented a higher content of epicatechin and procyanidin B1 when compared with control wines, whereas MeJ+Ur wines showed a higher content of epigallocatechin, procyanidin B1, and total flavanols in comparison with control wines. Differences among treatments in flavanols were not found.

**Table 3.** Flavonol, flavanol, phenolic acid and stilbene content (mg/L) in wines from control, methyl jasmonate (MeJ) and MeJ+Urea (MeJ+Ur) treatments, in seasons from 2019 and 2020.

| | 2019 | | | 2020 | | |
| | Control | MeJ | MeJ+Ur | Control | MeJ | MeJ+Ur |
|---|---|---|---|---|---|---|
| Flavonols | | | | | | |
| Myricetin-3-glcU | 12.16 ± 1.20 a | 10.40 ± 1.63 a | 12.29 ± 1.28 a | 6.64 ± 0.39 a | 6.71 ± 0.62 a | 7.94 ± 1.24 a |
| Myricetin-3-gal | 15.56 ± 0.34 b | 13.33 ± 1.19 a | 13.81 ± 0.85 a | 8.14 ± 1.05 a | 9.49 ± 1.06 a | 13.33 ± 1.72 b |
| Myricetin-3-glc | 110.56 ± 6.68 a | 105.43 ± 17.27 a | 119.18 ± 6.25 a | 31.94 ± 6.38 a | 47.86 ± 5.78 a | 65.18 ± 11.97 b |
| Quercetin-3-glcU | 85.40 ± 11.76 b | 60.07 ± 6.79 a | 53.31 ± 9.82 a | 11.35 ± 1.11 a | 13.12 ± 1.76 a | 14.89 ± 2.64 a |
| Quercetin-3-glc | 94.97 ± 11.20 b | 74.64 ± 6.63 a | 75.28 ± 8.84 a | 57.77 ± 6.23 a | 76.74 ± 9.28 b | 83.88 ± 10.62 b |
| Laricitrin-3-glc | 17.50 ± 1.22 a | 15.95 ± 1.78 a | 17.03 ± 0.62 a | 10.79 ± 0.37 a | 11.79 ± 1.22 a | 15.16 ± 2.29 b |
| Kaempferol-3-gal | 1.58 ± 0.23 b | 1.30 ± 0.23 ab | 1.12 ± 0.19 a | 0.16 ± 0.01 a | 0.19 ± 0.03 a | 0.18 ± 0.02 a |
| Kaempferol-3-glcU+3-glc | 7.24 ± 1.14 b | 4.95 ± 0.61 a | 5.09 ± 0.88 a | 0.70 ± 0.10 a | 0.78 ± 0.07 a | 0.99 ± 0.10 b |
| Isorhamnetin-3-glc | 1.73 ± 0.24 a | 1.66 ± 0.28 a | 1.76 ± 0.13 a | 0.23 ± 0.04 a | 0.38 ± 0.04 b | 0.60 ± 0.06 c |
| Syringetin-3-glc | 11.25 ± 1.06 a | 10.67 ± 1.73 a | 11.46 ± 0.51 a | 8.92 ± 0.59 a | 10.40 ± 1.24 ab | 12.92 ± 2.90 b |
| Free-myricetin | 12.56 ± 0.46 a | 15.85 ± 2.44 a | 13.38 ± 1.24 a | 18.61 ± 3.15 a | 30.71 ± 5.01 b | 25.61 ± 3.85 ab |
| Free-quercetin | 18.85 ± 1.69 b | 18.73 ± 3.00 b | 11.57 ± 2.19 a | 14.36 ± 1.39 a | 17.09 ± 2.46 ab | 19.22 ± 2.32 b |
| Free-kaempferol | 10.09 ± 0.69 b | 11.42 ± 1.48 b | 7.56 ± 0.60 a | 3.95 ± 0.32 a | 3.93 ± 0.09 a | 4.70 ± 0.38 b |
| Free-laricitrin | 2.34 ± 0.06 a | 2.36 ± 0.22 a | 2.08 ± 0.11 a | 4.70 ± 0.29 a | 5.37 ± 1.12 a | 5.67 ± 0.51 a |
| Free-isorhamnetin+syringetin | 0.54 ± 0.05 ab | 0.64 ± 0.07 b | 0.48 ± 0.08 a | 0.38 ± 0.03 a | 0.40 ± 0.05 a | 0.47 ± 0.10 a |
| Total flavonols | 402.34 ± 29.87 a | 343.84 ± 40.47 a | 361.02 ± 38.48 a | 178.57 ± 6.30 a | 225.67 ± 55.20 ab | 277.97 ± 53.04 b |
| Flavanols | | | | | | |
| Catechin | 16.62 ± 1.12 a | 18.37 ± 2.85 a | 20.49 ± 3.04 a | 8.18 ± 1.57 a | 8.17 ± 1.05 a | 8.90 ± 1.07 a |
| Epicatechin | 19.02 ± 1.22 ab | 18.49 ± 3.53 a | 23.16 ± 0.58 b | 10.07 ± 1.46 a | 14.32 ± 2.04 b | 11.76 ± 1.44 ab |
| Epicatechin-3-gallate | 17.24 ± 1.84 a | 16.71 ± 3.22 a | 18.78 ± 3.05 a | n.d. | n.d. | n.d. |
| Epigallocatechin | 1.50 ± 0.23 a | 2.32 ± 0.37 b | 1.98 ± 0.32 ab | 6.14 ± 0.93 a | 7.45 ± 0.73 a | 10.44 ± 1.82 b |
| Procyanidin B1 | 7.47 ± 0.96 a | 15.93 ± 1.11 c | 12.23 ± 1.36 b | 2.64 ± 0.42 a | 4.46 ± 0.57 b | 5.03 ± 1.00 b |
| Procyanidin B2 | 16.34 ± 1.50 b | 8.06 ± 1.53 a | 24.58 ± 3.75 c | n.d. | n.d. | n.d. |
| Total flavanols | 81.99 ± 2.40 a | 87.77 ± 16.59 a | 96.98 ± 9.19 a | 26.13 ± 4.77 a | 35.72 ± 3.47 ab | 36.91 ± 6.16 b |
| Hydroxybenzoic acid | | | | | | |
| Gallic acid | 29.84 ± 4.11 b | 20.17 ± 2.87 a | 29.69 ± 5.74 b | 14.46 ± 1.04 a | 18.89 ± 1.26 a | 18.10 ± 3.73 a |
| Hydroxycinnamic acids (HCAs) | | | | | | |
| *trans*-Caftaric acid | 4.42 ± 0.53 b | 2.27 ± 0.51 a | 7.48 ± 0.46 c | 9.19 ± 1.00 a | 12.23 ± 1.04 b | 9.88 ± 1.52 ab |
| *trans*+*cis*-Coutaric acids | 2.65 ± 0.29 b | 1.70 ± 0.32 a | 4.70 ± 0.22 c | 7.07 ± 0.71 a | 8.98 ± 0.83 a | 7.04 ± 1.36 a |
| *trans*-Fertaric acid | 1.12 ± 0.10 a | 0.93 ± 0.14 a | 1.07 ± 0.09 a | 1.48 ± 0.04 a | 1.90 ± 0.28 b | 2.06 ± 0.21 b |
| Caffeic acid | 30.43 ± 0.71 b | 22.49 ± 2.48 a | 30.02 ± 0.61 b | 12.11 ± 2.28 a | 14.50 ± 3.05 a | 15.42 ± 3.01 a |
| p-Coumaric acid | 10.52 ± 0.98 ab | 7.95 ± 0.10 a | 12.38 ± 2.07 b | 7.30 ± 1.46 a | 8.35 ± 1.55 ab | 11.45 ± 2.08 b |
| Ferulic acid | 2.31 ± 0.29 b | 1.83 ± 0.31 ab | 1.71 ± 0.16 a | 2.08 ± 0.37 a | 2.63 ± 0.30 ab | 3.14 ± 0.40 b |
| Total HCAs | 52.19 ± 3.53 a | 43.97 ± 10.35 a | 54.01 ± 7.90 a | 39.24 ± 2.48 a | 48.36 ± 3.65 ab | 50.09 ± 6.85 b |
| Stilbenes | | | | | | |
| *trans*-Piceid | 3.55 ± 0.22 a | 3.43 ± 0.56 a | 3.83 ± 0.20 a | 0.87 ± 0.08 a | 1.56 ± 0.20 b | 2.10 ± 0.41 c |
| *cis*-Piceid | 0.24 ± 0.04 a | 0.47 ± 0.06 b | 0.41 ± 0.03 b | 0.95 ± 0.13 a | 0.87 ± 0.09 a | 1.35 ± 0.16 b |
| *trans*-Resveratrol | 0.58 ± 0.02 a | 0.74 ± 0.12 b | 0.47 ± 0.02 a | 1.87 ± 0.07 a | 2.96 ± 0.22 b | 3.16 ± 0.38 b |
| *cis*-Resveratrol | 0.63 ± 0.10 a | 0.67 ± 0.06 a | 0.83 ± 0.02 b | 0.50 ± 0.04 a | 0.73 ± 0.15 ab | 0.77 ± 0.16 b |
| Total stilbenes | 5.15 ± 0.43 a | 5.23 ± 1.11 a | 5.67 ± 0.37 a | 4.28 ± 0.37 a | 5.93 ± 0.91 ab | 7.65 ± 1.43 b |

Nomenclature abbreviations: glcU, glucuronide; gal, galactoside; glc, glucoside. All parameters are listed with their standard deviation (*n* = 3). For each season and compound, different letters indicate significant differences between the samples ($p \leq 0.05$). n.d.: not detected.

Gallic acid was the only hydroxybenzoic acid detected (Table 3). In the first season, MeJ wines presented a significant decrease in this acid when compared with control and MeJ+Ur wines. However, in 2020, foliar treatments did not affect the content of this compound in the wines. Previous studies about the effect of foliar application of MeJ [15] and urea [18] to vineyard described the same trend, MeJ and urea treatments did not affect gallic acid content in wines.

The main hydroxycinnamic acid found in wines was caffeic acid in all wines and in both seasons (Table 3). In 2019, MeJ wines showed a lower content of *trans*-caftaric, *trans*+*cis*-coutaric, and caffeic acids than control wines, whereas MeJ+Ur wines showed a higher content of *trans*-caftaric, and *trans*+*cis*-coutaric acids, and a lower content of ferulic acid in comparison with control wines. Differences in the total hydroxycinnamic acids content was not found in agreement with the described by Portu et al. [15] for MeJ foliar application and Portu et al. [18] for urea foliar treatment. In the second vintage, MeJ

wines showed a higher content of *trans*-caftaric and *trans*-fertaric acids when compared with control wines, whereas MeJ+Ur wines presented a higher content of *trans*-fertaric, *p*-coumaric and ferulic acids (Table 3). MeJ+Ur wines were characterized by the highest total hydroxycinnamic acids content, whereas MeJ wines showed an intermediate value and control wines showed the lowest total hydroxycinnamic acids content.

Regarding stilbenes, in the first season, the main stilbene was *trans*-piceid in all samples (Table 3). MeJ and MeJ+Ur wines underwent an increase in *cis*-piceid and *trans*-resveratrol content; MeJ+Ur wines also showed a higher amount of *cis*-resveratrol than control wines. However, regarding the total stilbenes content, differences among wines were not detected. In the second season, the most abundant stilbene was *trans*-resveratrol. MeJ wines presented a higher content of *trans*-piceid, *trans*-resveratrol, and an intermediate content of total stilbenes in comparison with control wines, whereas MeJ+Ur wines showed an increase in the concentration of all stilbenes and therefore in total stilbene content. Therefore, MeJ+Ur foliar application produced an increase in the biosynthesis of stilbenes, which has been reflected in the wines. This effect has been previously described in grapes by Portu et al. [15], after MeJ foliar application, and by Portu et al. [39], after urea foliar treatment.

Overall, the total content of the families of non-flavonoid compounds (hydroxycinnamic acids and stilbenes) in wines was not affected by foliar treatments when compared with control wines, in the first season (Table 3). This fact could be explained because non-flavonoid synthesis happens usually before veraison and foliar treatments were applied at origin and one week later, which makes sense seeing that the effect of foliar treatments in these compounds was minor [1], except for gallic acid, which underwent a decrease in MeJ wines in 2019. Nevertheless, in the second vintage (2020), MeJ+Ur foliar treatment produced a significant increase in total hydroxycinnamic acids and total stilbenes content (Table 3). Potentially, the combined application of MeJ plus urea could reduce the effect of application of MeJ on the increase in protein content in the skin cell wall that produced berries with a more rigid cell wall structure, as mentioned above.

### 3.3. Effect of the Foliar MeJ and MeJ+Ur Applications on Wine Aromatic Compounds

Esters, higher alcohols and acids are quantitatively dominant in wine aroma, and therefore have a high influence in the sensory properties and quality of wines [7]. The ester content in Tempranillo wines from control, MeJ and MeJ+Ur treatments in 2019 and 2020 is presented in Figure 1.

Isoamyl acetate and 2-phenylethyl acetate were the two acetate esters detected in the wines. In the first vintage, control wines showed the highest content of isoamyl acetate, 2-phenylethyl acetate and therefore, total acetate esters (Figure 1a–c). MeJ wines presented a higher content of 2-phenylethyl acetate than MeJ+Ur wines, but those wines did not show differences in total acetate ester content. In 2020, the effect of foliar treatments in wines was different. Control and MeJ+Ur wines did not show differences in this family of compounds. Nevertheless, MeJ wines were characterized by a lower content of isoamyl acetate, 2-phenylethyl acetate and total acetate esters when compared with MeJ+Ur wines (Figure 1a–c). Ancín-Azpilicueta et al. [24] showed a decrease in isoamyl acetate in urea wines with respect to the content of control wines, whereas Rubio-Bretón et al. [25] did not show differences in this compound between control and urea wines.

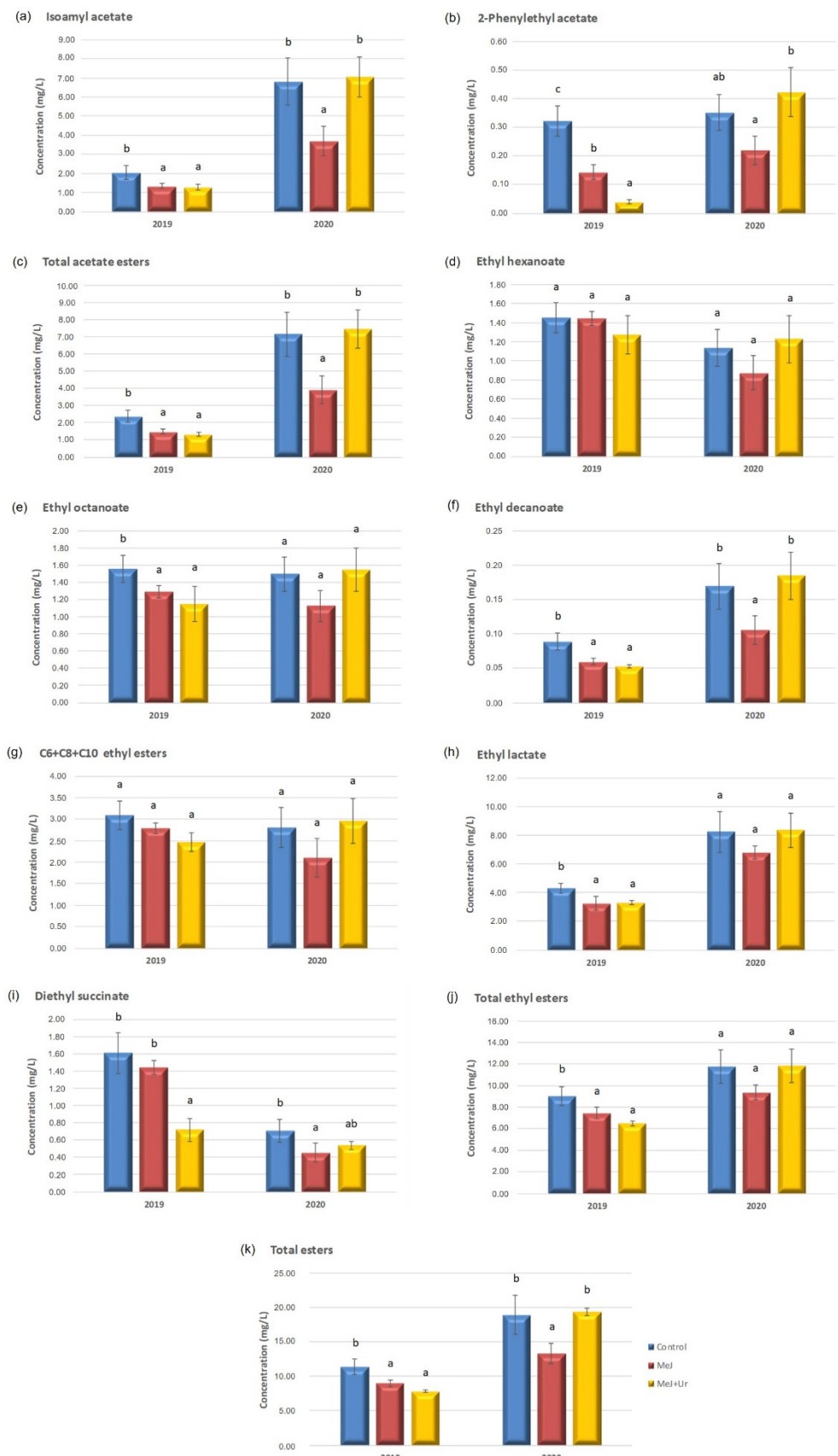

**Figure 1.** Ester concentration (mg/L) in Tempranillo wines from control, MeJ and MeJ+Ur treatments in 2019 and 2020 seasons. All parameters are given with their standard deviation (*n* = 3). For each season and compound, different letters indicate significant differences between samples (*p* ≤ 0.05).

Figure 1d–j show the ethyl esters content in the wines. Ethyl esters of the fatty acids are a relevant group for wine aroma, especially from C6 and C10 acids; they are produced during yeast fermentation from the ethanolysis of acyl-CoA [13]. In 2019, MeJ and MeJ+Ur wines showed a lower content of ethyl octanoate and ethyl decanoate when compared with control wines, whereas differences in the content of ethyl hexanoate and C6+C8+C10 ethyl esters were not detected among the wines. Ethyl decanoate and ethyl octanoate realize a key role as aroma enhancer compounds [7]. MeJ and MeJ+Ur wines presented a lower content of ethyl lactate regarding control wines; ethyl lactate content in wines is directly related to the lactic acid produced during the malolactic fermentation [40]. MeJ+Ur wines also showed a lower content of diethyl succinate when compared with control and MeJ wines (Figure 1i). Both wines from foliar treatments showed lower total ethyl ester content in comparison with control wines (Figure 1j). Regarding total ester concentration (Figure 1k), foliar treatments produced wines with a significant lower total concentration in comparison with control wines. In the second vintage, MeJ+Ur foliar treatment influenced less of the ester content in comparison with control wines. MeJ wines showed a lower content of isoamyl acetate and total acetate esters when compared with control and MeJ+Ur wines. Wines did not show differences in 2-phenylethyl acetate content. Regarding the content of C6–C10 esters, the only difference was a decrease in ethyl octanoate in MeJ wines respect to control and MeJ+Ur wines. However, it did not affect the total sum of C6, C8 and C10 ethyl esters (Figure 1d–g). MeJ wines also showed a lower content of diethyl succinate in comparison with control wines, although MeJ wines did not show differences in this compound with MeJ+Ur wines (Figure 1i). In spite of these effects, differences among control, MeJ and MeJ+Ur wines in total ethyl esters were not found (Figure 1j). However, the decrease observed in MeJ wines in several esters produced a significant lower content of total esters in MeJ wines when compared with control and MeJ+Ur wines (Figure 1k).

Goméz-Plaza et al. [13] described a slight effect of MeJ in wine esters content: some esters increased their content when compared with control wines, but total ester content of wines was not affected by MeJ foliar application. Ancín-Azpilicueta et al. [24] concluded that an increase in the esters content of wines is only achieved when the must is poor in nitrogen, due to, when assimilable nitrogen is not limited, the amino acids are used for the cellular process and for this reason did not produce esters, whereas Rubio-Bretón et al. [25] described how foliar application of urea barely affect the ester content in the wines.

The alcohol and acid concentration in Tempranillo wines from control, MeJ and MeJ+Ur treatments, in 2019 and 2020 seasons, is shown in Figure 2. Higher alcohols can be formed catabolically from amino acids via the Ehrlich pathway or anabolically from sugars [5]. In the first season, MeJ and MeJ+Ur wines showed a lower content of isobutanol, isoamyl alcohols, 2-phenylethanol, methionol, (E)-3-hexenol, and total alcohols in comparison with the content of control wines, in all cases the content of these compounds being lowest in the MeJ+Ur wines. Differences in the n-hexanol content were not found among wines (Figure 2e). Therefore, foliar application of MeJ and MeJ+Ur did not seem to improve the biosynthesis of alcohols, which can be good for the aroma quality of wines, since Rapp and Mandery [41] have described that, at concentrations below 300 mg/L, these compounds can contribute to the desirable complexity of wine, but in higher concentrations, above 400 mg/L, these compounds have a negative influence on wine aroma. Fortunately, control wines and wines elaborated with grapes from grapevines treated presented a total concentration of alcohols below of 300 mg/L (Figure 2g), which is positive for wine quality. Furthermore, in the second season, slight differences in alcohols were detected among control, MeJ and MeJ+Ur wines. MeJ wines presented a higher content of methionol than MeJ+Ur wines, but both wines did not show differences with control wines (Figure 2d). MeJ and MeJ+Ur wines showed a lower content of (E)-3-hexenol when compared with control wines (Figure 2f). Moreover, differences in total alcohol content were not found (Figure 2g). Rubio-Bretón et al. [25], in their study about different nitrogen foliar applications to vineyard, concluded that the content of alcohols in wines was scarcely affected by urea foliar application, whereas Ancín-Azpilicueta et al. [24] described a decrease in

the total concentration of higher alcohols in wines elaborated with grapes from vines treated with urea, as was observed in MeJ+Ur wines, in the first season. On the other hand, Gómez-Plaza et al. [13] showed an increase in certain alcohols and in the total alcohols content in MeJ wines in contrast with the results described in this work in both seasons studied (Figure 2). In the same way, D'Onofrio et al. [14] also showed an increase in several alcohols and in the total alcohols content of wines elaborated with grapes treated with MeJ.

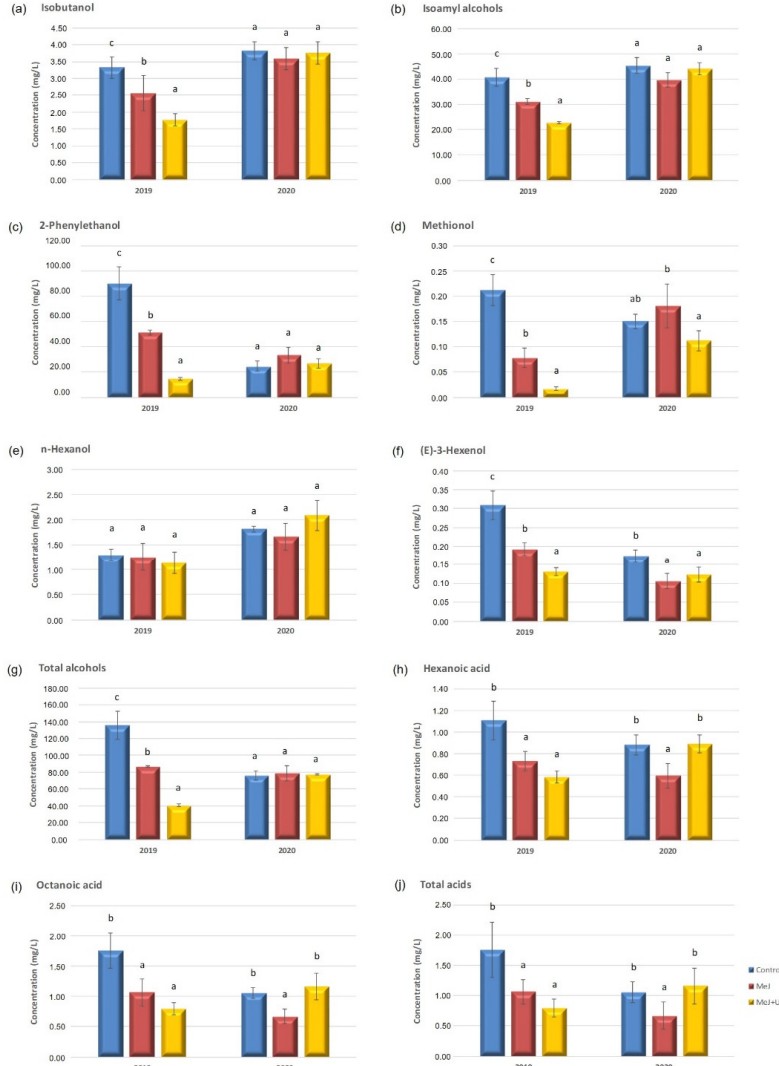

**Figure 2.** Alcohols and acids concentration (mg/L) in Tempranillo wines from control, MeJ and MeJ+Ur treatments in 2019 and 2020 seasons. All parameters are given with their standard deviation (*n* = 3). For each season and compound, different letters indicate significant differences between samples ($p \leq 0.05$).

Regarding acids (Figure 2h–j), in the first season, both foliar treatments produced wines with a lower content of hexanoic and octanoic acids and therefore, a lower total content of acids. In the second vintage, MeJ wines showed a lower content of hexanoic and octanoic acids and total acids comparing with control and MeJ+Ur wines. The synthesis of C6 and C8 acids is performed by yeasts, and it is related to the metabolism of carbohydrates, since glucose is the principal source of acetyl-CoA [8]. These compounds are related to a fresh flavor in wine, but in high concentrations (20 mg/L) they are associated with an unpleasant flavor [40]. The total acid content in all wines of this study was below 20 mg/L (Figure 2j); thus, these compounds should give a fresh aroma to control, MeJ and MeJ+Ur wines. Ancín-Azpilicueta et al. [24] concluded that the content of hexanoic and octanoic

acids in wines was not affected by urea application, whereas Rubio-Bretón et al. [25] described a slight effect of urea foliar application in the content of acids in wines, since wines from grapes treated with the low dose of urea presented a decrease in hexanoic acid content. Regarding the effect of MeJ application to the vineyard, Gómez-Plaza et al. [13] described that the wines obtained from the treated grapes did not show differences with control wines in acids. However, this result contrast with that observed in this work, since MeJ wines presented a decrease in the content of acids, in both seasons studied (Figure 2).

### 3.4. Influence of the Foliar MeJ and MeJ+Ur Treatments on Wine Nitrogen Compounds

Amino acid content in wines from control, methyl jasmonate (MeJ) and MeJ+Urea (MeJ+Ur) treatments, from both seasons (2019 and 2020), is shown in Table 4. In the first season, MeJ wines showed slight differences with control wines, in spite of the fact that MeJ foliar treatments in 2019 increased the content of several amino acids in grapes (data pending publication). MeJ wines presented a higher content of aspartic acid, asparagine, leucine, phenylalanine, ornithine, lysine, and total amino acids without proline when compared with control wines. Proline is an amino acid that yeast cannot metabolized, in normal conditions, and for this reason, it is interesting to know the total amino acid content without proline. On the other hand, MeJ+Ur wines showed a higher content of all amino acids, except arginine and cysteine, when compared with control wines (Table 4). This result found in wines agrees with the observed in grapes, although MeJ+Ur grapes presented higher content of arginine and cysteine than control grapes (data pending publication).

Regarding total amino acids, MeJ+Ur wines showed a higher content when compared with control and MeJ wines. MeJ+Ur wines also presented a higher content of total amino acids without proline in comparison with control wines but did not show differences with MeJ wines (Table 4).

In the second vintage, MeJ wines presented a lower content of histidine, arginine, tyrosine, leucine, phenylalanine, ornithine, lysine, and total amino acids without proline when compared with control wines (Table 4). Furthermore, MeJ+Ur wines showed a lower content of aspartic acid, glutamic acid, γ-aminobutyric acid, tyrosine, valine, methionine, isoleucine+tryptophan, leucine, phenylalanine, ornithine, lysine, and total amino acids without proline in comparison with control wines. Wines from foliar treated vines presented differences, MeJ+Ur wines showed a higher content of arginine, proline, and total amino acids, and a lower content of aspartic acid, glutamic acid, γ-aminobutyric acid, tyrosine, valine, methionine, and isoleucine+tryptophan in comparison with MeJ wines (Table 4).

These effects in amino acids are diverse from those described in grapes (data pending publication), probably due to these nitrogen compounds are consumed by yeast during the alcoholic fermentation [23]. These results contrast with those described by Gil-Muñoz et al. [23] for Monastrell wines since these authors observed the effect of MeJ foliar application on wines.

**Table 4.** Amino acids content (mg/L) in wines from control, methyl jasmonate (MeJ) and MeJ+Urea (MeJ+Ur) treatments, in 2019 and 2020 seasons.

| | 2019 | | | 2020 | | |
| | Control | MeJ | MeJ+Ur | Control | MeJ | MeJ+Ur |
|---|---|---|---|---|---|---|
| Aspartic acid | 0.07 ± 0.02 a | 1.40 ± 0.46 b | 4.80 ± 0.15 c | 7.27 ± 0.65 b | 6.67 ± 0.56 b | 4.63 ± 1.29 a |
| Glutamic acid | 3.44 ± 0.88 a | 5.73 ± 2.25 a | 25.45 ± 4.96 b | 16.38 ± 1.32 b | 17.88 ± 4.51 b | 10.46 ± 1.74 a |
| Asparagine | 3.36 ± 0.82 a | 5.78 ± 1.23 b | 15.01 ± 0.09 c | 8.22 ± 1.31 a | 7.62 ± 1.12 a | 6.79 ± 1.50 a |
| Serine | 3.17 ± 1.01 a | 3.13 ± 1.49 a | 6.72 ± 0.06 b | 7.71 ± 1.11 a | 7.44 ± 1.00 a | 5.96 ± 0.66 a |
| Glutamine | 2.79 ± 0.20 a | 1.99 ± 0.95 a | 16.05 ± 2.28 b | 6.89 ± 1.01 a | 5.27 ± 1.33 a | 6.85 ± 1.77 a |
| Histidine | 5.25 ± 1.10 a | 5.51 ± 1.44 a | 12.75 ± 2.04 b | 13.10 ± 2.38 b | 7.83 ± 1.43 a | 10.88 ± 1.57 ab |
| Glycine | 6.48 ± 0.41 a | 8.85 ± 1.89 a | 19.96 ± 1.52 b | 15.29 ± 2.04 a | 14.80 ± 3.26 a | 12.48 ± 1.18 a |
| Threonine+Citrulline | 1.82 ± 0.22 a | 3.46 ± 1.13 a | 23.84 ± 4.80 b | 10.62 ± 1.23 a | 8.12 ± 1.82 a | 8.61 ± 0.88 a |
| Arginine | 6.02 ± 0.28 a | 6.51 ± 0.36 a | 5.65 ± 0.86 a | 7.09 ± 1.85 b | 4.34 ± 0.69 a | 8.09 ± 0.87 b |
| Alanine | 3.52 ± 0.99 a | 8.05 ± 3.37 a | 23.08 ± 5.78 b | 26.21 ± 5.20 a | 21.56 ± 2.53 a | 18.76 ± 2.64 a |
| γ-Aminobutyric acid | 9.06 ± 1.37 a | 16.69 ± 5.17 a | 117.24 ± 9.92 b | 14.24 ± 1.83 b | 15.17 ± 2.61 b | 6.29 ± 0.17 a |
| Proline | 647.05 ± 45.92 a | 726.77 ± 110.61 a | 985.59 ± 96.06 b | 2172.04 ± 120.58 ab | 1816.80 ± 218.65 a | 2397.73 ± 272.59 b |
| Tyrosine | 0.63 ± 0.05 a | 1.96 ± 1.72 a | 4.59 ± 1.18 b | 6.68 ± 0.67 c | 4.94 ± 0.80 b | 3.19 ± 0.15 a |

**Table 4.** *Cont.*

| | | 2019 | | | 2020 | |
| --- | --- | --- | --- | --- | --- | --- |
| | **Control** | **MeJ** | **MeJ+Ur** | **Control** | **MeJ** | **MeJ+Ur** |
| Valine | $0.67 \pm 0.06$ a | $2.32 \pm 1.72$ ab | $3.02 \pm 0.24$ b | $7.46 \pm 0.96$ b | $6.34 \pm 0.87$ b | $4.22 \pm 0.45$ a |
| Methionine | $0.56 \pm 0.10$ a | $0.86 \pm 0.47$ a | $1.53 \pm 0.26$ b | $1.69 \pm 0.28$ b | $1.39 \pm 0.28$ b | $0.58 \pm 0.11$ a |
| Cysteine | $0.44 \pm 0.06$ b | $0.27 \pm 0.06$ a | $0.31 \pm 0.09$ ab | $0.36 \pm 0.04$ a | $0.38 \pm 0.06$ a | $0.39 \pm 0.09$ a |
| Isoleucine+Tryptophan | $0.93 \pm 0.07$ a | $1.87 \pm 0.96$ a | $4.72 \pm 0.27$ b | $7.60 \pm 0.76$ b | $7.25 \pm 0.90$ b | $4.59 \pm 0.50$ a |
| Leucine | $1.40 \pm 0.32$ a | $4.84 \pm 1.00$ b | $4.86 \pm 0.56$ b | $12.94 \pm 2.44$ b | $7.72 \pm 0.90$ a | $5.14 \pm 0.44$ a |
| Phenylalanine | $0.94 \pm 0.19$ a | $2.77 \pm 0.68$ b | $3.29 \pm 0.26$ b | $9.52 \pm 1.49$ b | $5.66 \pm 0.72$ a | $4.58 \pm 0.39$ a |
| Ornithine | $3.26 \pm 0.25$ a | $49.27 \pm 23.18$ b | $54.72 \pm 3.28$ b | $33.74 \pm 3.30$ b | $16.98 \pm 1.40$ a | $21.96 \pm 3.37$ a |
| Lysine | $2.42 \pm 0.40$ a | $8.03 \pm 0.87$ b | $7.39 \pm 0.36$ b | $26.88 \pm 3.35$ b | $19.89 \pm 3.20$ a | $15.34 \pm 0.40$ a |
| Total AA | $703.27 \pm 41.79$ a | $866.04 \pm 121.48$ a | $1340.55 \pm 106.41$ b | $2411.95 \pm 135.64$ ab | $2004.05 \pm 237.31$ a | $2557.51 \pm 285.93$ b |
| Total AA without Pro | $56.22 \pm 7.14$ a | $139.27 \pm 27.70$ b | $357.97 \pm 19.64$ b | $239.91 \pm 23.20$ b | $187.25 \pm 21.38$ a | $159.78 \pm 13.78$ a |

Total AA: Total amino acids. All parameters are listed with their standard deviation ($n = 3$). For each season and compound, different letters indicate significant differences between the samples ($p \leq 0.05$).

## 4. Conclusions

Phenolic, aromatic and nitrogen compounds are key to quality of wine. This work studied for first time the effect of MeJ and MeJ+Ur foliar application over phenolic, volatile and amino acid content in Tempranillo wines over two consecutive seasons. Foliar treatments applied to vineyard produced slight effect on enological parameters in wines. The reduction in alcoholic degree observed in the first vintage had a more interesting impact on the reduction in the effects of climatic change, although this effect was season-dependent. With respect to anthocyanin content, MeJ wines presented a higher content of total acylated in comparison with control wines, but did not show differences with MeJ+Ur wines. This increased interest for wine ageing. Overall, foliar treatments slightly affected wine phenolic content, but only in 2020; MeJ+Ur wines presented a higher content of total flavonols, flavanols, hydroxycinnamic acids, and stilbenes than control wines. In the first season, MeJ and MeJ+Ur wines showed a lower content of total esters, total alcohols and total acids when compared with control wines, whereas in the second vintage, MeJ wines presented lower content of total esters and acids; however, MeJ+Ur wines did not show differences in total content of the different families of volatile compounds studied when compared with control wines. Finally, with regard to amino acids, MeJ+Ur wines showed a higher total amino acids content than control and MeJ wines in the first season, whereas in 2020, MeJ+Ur wines showed a higher total amino acids content than MeJ wines but did not show difference with control wines. The different effect of foliar applications to vineyard on phenolic, volatile and nitrogen compounds in wines between seasons confirms that these treatments were season-dependent. The slight effect of MeJ and MeJ+Ur foliar application with regard to phenolic compounds indicates that the winemaking process must be carried out thoroughly to extract grape compounds to wines. The combinate foliar application of MeJ and urea more greatly enhanced the phenolic and nitrogen content and slightly affected the volatile composition of wines in comparison with the MeJ foliar treatment.

**Supplementary Materials:** The following supporting information can be downloaded at: https://www.mdpi.com/article/10.3390/beverages8030052/s1, Table S1: Climatic conditions during all vegetative cycle of both vintages.

**Author Contributions:** Conceptualization, E.P.P.-Á., P.R.-B. and T.G.-C.; methodology, E.P.P.-Á., P.R.-B. and T.G.-C.; formal analysis, I.S.d.U., R.M.-P. and S.M.-S.R.; investigation, M.G.-L., E.P.P.-Á., P.R.-B. and T.G.-C.; resources, E.P.P.-Á., P.R.-B. and T.G.-C.; data curation, M.G.-L., I.S.d.U. and T.G.-C.; writing—original draft preparation, M.G.-L.; writing—review and editing, I.S.d.U., R.M.-P., S.M.-S.R., E.P.P.-Á., P.R.-B. and T.G.-C.; funding acquisition, E.P.P.-Á., P.R.-B. and T.G.-C. All authors have read and agreed to the published version of the manuscript.

**Funding:** This work has been carried out thanks to funding from the Ministerio de Ciencia, Innovación y Universidades through the Project RTI2018-096549-B-I00.

**Institutional Review Board Statement:** Not applicable.

**Informed Consent Statement:** Not applicable.

**Data Availability Statement:** Not applicable.

**Acknowledgments:** M.G.-L. thanks to the Universidad de La Rioja for her Margarita Salas contract funding by the Ministerio de Universidades and the European Union (Financed by the European Union-Next GenerationEU). R.M.-P. thanks to INIA for her predoctoral contract. S.M.-S.R. thanks Gobierno de La Rioja for her predoctoral contract. E.P.P.-Á. thanks the Ministerio de Ciencia, Innovación y Universidades for her Juan de la Cierva-Incorporación contract.

**Conflicts of Interest:** The authors declare no conflict of interest.

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
