# Peer review of "Effect of Methyl Jasmonate and Methyl Jasmonate Plus Urea Foliar Applications on Wine Phenolic, Aromatic and Nitrogen Composition"

_beverages, doi:10.3390/beverages8030052_

Round 1

Reviewer 1 Report

According to authors all analysis methods (HPLC , GC)are already available in literature sources. Never the less this work is missing information how all methods were tested to ensure proper results are obtained. At least basic method transfer is needed to check specificity, range, repeatabilitylinearity etc. I believe such tests were performed but not described. Please include information how literature methods performance in this specific use were evaluated. This is especially important for HPLC and GC methods. A summary of these activities should be included. 

Specific Comments:

Line 167 - please indicate which standards were commercial and from which company

Line 171 no validation of the HPLC method (selectivity, precision, repeatability, linearity etc.)

Line 194 no validation of the GC method (selectivity, precision, repeatability, linearity etc.)

Line 215 please indicate which standards were commercial and from which company

Line 218 no validation of the HPLC method (selectivity, precision, repeatability, linearity etc.)

Author Response

MANUSCRIPT beverages-1831773

 RESPONSE TO REVIEWER 1 AND CHANGES WE HAVE MADE TO THE MANUSCRIPT

Firstly, please accept our thanks for the comments you have made on our paper. Your suggestions helped us to improve the quality of the paper. We hope that the answers outlined below and the changes we have made to the text will be satisfactory.

Line 167- please indicate which standards were commercial and from which company

 Response: Information completed in the text: “Hence, malvidin-3-O-glucoside (Extrasynthèse, Genay, France) was used for anthocyanins, quercetin-3-O-glucoside (Sigma-Aldrich) was used for flavonols, gallic acid was quantified with gallic acid (Sigma-Aldrich), trans-caftaric acid (Extrasynthèse) was used for free hydroxycinnamic acids and the corresponding tartaric esters, catechin (Sigma-Aldrich) was used for procyanidins B1 and B2, epicatechin (Sigma-Aldrich) was used for epigallocatechin, and trans-piceid (Sigma-Aldrich) and trans-resveratrol (Sigma-Aldrich) were used for their respective cis isomers. Concentrations of phenolic compounds in wines were expressed as mg/L.”

Line 171, no validation of the HPLC method (selectivity, precision, repeatability, linearity etc.)

Response: Information added in the text: “The characteristics of the HPLC method were the following: variation coefficient (%) for retention time of commercially standards varied from 0.09 to 0.72; the detection limit (mg/L) ranged from 0.099 to 0.711; the quantification limit (mg/L) changed from 0.292 to 2.370; the variation coefficient (%) for concentration varied from 1.66 to 6.67. The response factor (mg/area units) was also calculated ranging from 3.99E-06 to 1.00E-4. The variation coefficients were obtained from 10 consecutive analyses.”

Line 194, no validation of the GC method (selectivity, precision, repeatability, linearity etc.)

Response: Information added in the text: “For the validation of the GC-MS method were obtained these characteristics: limit of quantification (µg/L) was ranged from 1.8 to 107.4; the concentration range (µg/L) varied from 2.48 to 2132; precision was expressed as relative standard deviation (RSD, %) and changed from 5.7 to 19.7; the repeatability (%) was measured from 2.6 to 9.0; the accuracy was expressed as relative error (%) and varied from 0.7 to 20.5.”

Line 215, please indicate which standards were commercial and from which company

Response: We have all the standards for amino acids analyses, the information has been clarified in the text: “(aspartic acid, glutamic acid, asparagine, serine, glutamine, histidine, glycine, threonine, citrulline, arginine, alanine, γ-aminobutyric acid, proline, tyrosine, valine, methionine, cysteine, isoleucine, tryptophan, leucine, phenylalanine, ornithine, and lysine, all from Sigma-Aldrich)”

Line 218, no validation of the HPLC method (selectivity, precision, repeatability, linearity etc.)

Response: This information has been included in the text: “The characteristics measured in the validation of the method were: range of calibration (mg/L) from 1.00 to 2484; the repeatability of the method was studied and the resulting variation coefficients were below 5 %; and the detection limits for the amino acids were below 0.4 mg/L.”

Reviewer 2 Report

In their Manuscript « beverages-1831773» entitled " Effect of methyl jasmonate and methyl jasmonate plus urea foliar applications on wine phenolic, aromatic and nitrogen composition" for Beverages (journal), authors have reported somewhat original data and I think that the study is very interesting and it deserves to be published.

However, some minor points of criticism and questions have to be clarified prior publication:

MINOR REVISION:

Table 1: Last parameters is TPI (and not TP); typing mistake.

Lines 321, 338, 368…: I would suggest to include, as Supplementary Material, data about climatic conditions (rainfalls, temperatures…) during all vegetative cycle. This supplementary material should be linked with observed differences in results between 2019 and 2020 harvestings.

Line 372: paper “Preharvest Application of Elicitors to Monastrell Grapes: Impact on Wine Polysaccharide and Oligosaccharide Composition” by Apolinar-Valiente et al. (2018) could help to explain with greater depth this behavior.

Table 4 and line 562: Why “Total AA without proline” parameter is included? What is its meaning? This point must be clarified.

Line 573: “…wine’s quality…” (typing mistake)

Author Response

MANUSCRIPT beverages-1831773

 RESPONSE TO REVIEWER 2 AND CHANGES WE HAVE MADE TO THE MANUSCRIPT

 We would like to thank the reviewer for his/her useful comments and appreciation of our paper. All the points raised are answered each by each and the relevant parts of the manuscript are modified accordingly. Besides, the manuscript has been modified with comments from the others reviewers.

Table 1: Last parameters is TPI (and not TP); typing mistake.

Response: It has been corrected.

Lines 321, 338, 368…: I would suggest to include, as Supplementary Material, data about climatic conditions (rainfalls, temperatures…) during all vegetative cycle. This supplementary material should be linked with observed differences in results between 2019 and 2020 harvestings.

Response: We have elaborated a supplementary table with meteorological data (Table 1S) and, we have added in the text a short explanation about climatic conditions (Lines 340, 359-361, 383-388).

Line 372: paper “Preharvest Application of Elicitors to Monastrell Grapes: Impact on Wine Polysaccharide and Oligosaccharide Composition” by Apolinar-Valiente et al. (2018) could help to explain with greater depth this behavior.

Response: The reference has been included and we have added the information about the reinforce of skin cell wall of grapes due to the MeJ treatment (Lines: 385-388).

Table 4 and line 562: Why “Total AA without proline” parameter is included? What is its meaning? This point must be clarified.

Response: Information in the text has been added to clarify this point. “Proline is an amino acid that yeast cannot metabolized, in normal conditions, and for this reason, it is interesting to know the total amino acid content without proline.” (Lines 563-565).

Line 573: “…wine’s quality…” (typing mistake)

Response: This typing mistake has been corrected. Wine’s quality has been changed by quality of wine (line 595).
